# High Throughput Small Molecule Screen for Reactivation of *FMR1* in Fragile X Syndrome Human Neural Cells

**DOI:** 10.3390/cells11010069

**Published:** 2021-12-27

**Authors:** Jack F. V. Hunt, Meng Li, Ryan Risgaard, Gene E. Ananiev, Scott Wildman, Fan Zhang, Tim S. Bugni, Xinyu Zhao, Anita Bhattacharyya

**Affiliations:** 1Waisman Center, University of Wisconsin-Madison, Madison, WI 53705, USA; jackfaro@gmail.com (J.F.V.H.); limeng@xzhmu.edu.cn (M.L.); rrisgaard@wisc.edu (R.R.); 2Department of Neuroscience, School of Medicine and Public Health, University of Wisconsin-Madison, Madison, WI 53705, USA; 3Carbone Cancer Center Drug Discovery Core, University of Wisconsin-Madison, Madison, WI 53705, USA; geananiev@wisc.edu (G.E.A.); swildman@wisc.edu (S.W.); 4School of Pharmacy, University of Wisconsin-Madison, Madison, WI 53705, USA; fzhang83@wisc.edu (F.Z.); tim.bugni@wisc.edu (T.S.B.); 5Department of Cell and Regenerative Biology, School of Medicine and Public Health, University of Wisconsin-Madison, Madison, WI 53705, USA

**Keywords:** fragile X syndrome, drug screen, human pluripotent stem cells, *FMR1*, gene reactivation, small molecule

## Abstract

Fragile X syndrome (FXS) is the most common inherited cause of autism and intellectual disability. The majority of FXS cases are caused by transcriptional repression of the *FMR1* gene due to epigenetic changes that are not recapitulated in current animal disease models. FXS patient induced pluripotent stem cell (iPSC)-derived gene edited reporter cell lines enable novel strategies to discover reactivators of *FMR1* expression in human cells on a much larger scale than previously possible. Here, we describe the workflow using FXS iPSC-derived neural cell lines to conduct a massive, unbiased screen for small molecule activators of the *FMR1* gene. The proof-of-principle methodology demonstrates the utility of human stem-cell-based methodology for the untargeted discovery of reactivators of the human *FMR1* gene that can be applied to other diseases.

## 1. Introduction

Fragile X syndrome (FXS) is the most common inherited cause of autism spectrum disorder and intellectual disability, affecting an estimated 1 in 5000 males in the general population [1]. The neuropsychiatric symptoms and sequelae of FXS, including social anxiety, hyperactivity, developmental delay, autism symptoms, and seizures, typically cause the greatest medical, social, and financial burden for affected individuals and families [2,3]. The neuropsychiatric endophenotype of FXS is attributable to the loss of expression of the *FMR1* gene during development. In almost all cases, this loss of expression is due to an expansion of a CGG trinucleotide repeat region in the 5′ UTR of the *FMR1* gene to >200 copies. This expanded CGG region subsequently leads to CpG hypermethylation, histone modification, and transcriptional repression of the *FMR1* gene [4,5,6]. *FMR1* encodes the FMRP translational regulator 1 protein (also called FMRP), an RNA-binding protein that is highly expressed in the brain [7] and canonically functions by regulating the translation of a large number of target genes [8].

There are currently no specific pharmacological interventions for FXS. Preclinical work using FXS murine models has yielded valuable insight into the dysfunction of specific cellular processes associated with FMRP loss [9,10,11,12] and has largely guided interventional strategies up to this point. However, despite promising preclinical results in FXS animal models [13,14,15], clinical trials have shown modest results or failed to meet primary outcome measures [16,17,18,19,20,21,22,23,24,25,26]. The mixed results of these trials underscore an opportunity for new approaches to identify novel treatment targets for FXS [17].

Human cell-based strategies for treatment target discovery address some of the limitations of traditional animal model-based approaches [27,28]. Human induced pluripotent stem cells (iPSCs) recapitulate the hallmark epigenetic pathogenesis of FXS, harboring the full FXS CGG expansion mutation and exhibiting *FMR1* CpG hypermethylation and drastically reduced *FMR1* mRNA expression. Neural progenitor cells (NPCs) and neurons differentiated from FXS patient-derived cells retain their CpG methylation status and reduced *FMR1* mRNA [29,30].

Targeted screens for *FMR1*-reactivating treatments using FXS iPSCs and derived NPCs have confirmed or identified de-novo several small molecules capable of increasing *FMR1* expression in mitotic cells [31,32,33]. Molecules capable of reactivating *FMR1* expression, including DNA methyltransferase inhibitors such as 5-azacytidine and its derivatives [34], a histone deacetylase inhibitor [31], and a histone methyltransferase inhibitor (3-deazaneplanocin) [32], were identified from these and previous studies. However, these pharmacological agents all share a common mechanism of action based on blockade of DNA methylation or deacetylation during cell replication. The efficacy of these agents is cell cycle-dependent, with the major effects occurring only during the S phase [35,36]. Small molecules capable of reactivating *FMR1* expression in post-mitotic neurons, the primary target cell type responsible for FXS symptomatology, will likely demonstrate an alternative mechanism of action and will be required for inclusion in future targeted screening studies.

Many iPSC drug screens rely on mRNA, protein, or morphological analysis as readouts that are often characterized by low reproducibility and high variability and are time-consuming. The identification of compounds that reactivate *FMR1* in iPSC-derived neural progenitors have relied on antibodies to detect FMRP, which are limited in signal-to-background ratio, time-consuming, and expensive [31,33]. The use of a direct reporter line provides a more robust, simpler, and more economical assay that can be readily adapted to high throughput screening.

Recently, Nano luciferase *FMR1* (*FMR1*-Nluc) reporter iPSC lines from FXS patients were generated using CRISPR/Cas9 gene-editing strategies [30,37]. These reporter iPSCs can be differentiated into neural lineage cells, provide a highly sensitive readout of *FMR1* expression, and can be scaled to test thousands of candidate treatments simultaneously [30,38], making them an ideal tool to perform large, unbiased screens for novel reactivators of *FMR1* expression. In the present study, we used an FXS *FMR1* reporter line [30,37] to optimize and conduct an untargeted screen of over 320,000 small molecules for *FMR1* reactivation (Figure 1). We demonstrate the utility of human stem cell-based platforms for drug discovery by describing the workflow for the largest unbiased molecular screen for the reactivation of the *FMR1* gene published to date.

## 2. Materials and Methods

### 2.1. FMR1-Nano Luciferase Reporter Cell Lines

Human FXS iPSC stem cell lines (FXS-iPSCs) used to establish *FMR1* reporter lines have been described previously (FX13-2, >435 CGG repeats) [29]. The generation of the *FMR1* P2A-Nano luciferase (P2A-Nluc) reporter line has been previously described in full [30,37] using the same strategy as described [39]. Briefly, using CRISPR/Cas9 genome editing, a P2A-Nluc gene from the pNL1.1 template plasmid (Promega, Madison, WI, USA) was inserted at the endogenous *FMR1* locus of FXS-iPSCs, downstream of the CGG expansion region. The reporter line maintains the CGG repeats, methylation status, and *FMR1* expression status of the parental line [30]. FX-iPSC-Nluc1 iPSCs were maintained on a MEF feeder layer (WiCell) in an hESC medium of DMEM/F12 (Thermo Fisher, Waltham, MA, USA), 20% KnockOut™ serum replacement (Thermo Fisher, Waltham, MA, USA), 1 mM L-glutamine (Thermo Fisher, Waltham, MA, USA), 100 μm 2-mercaptoethanol (Sigma, St. Louis, MO USA), and 4 ng/mL human recombinant FGF-2 (Waisman Biomanufacturing), changed daily.

### 2.2. iPSC Neural Differentiation

For differentiation of iPSCs into neural progenitor cells, an established dual SMAD inhibition method [40] was used as previously modified [30]. Reporter NPCs were maintained on Matrigel (Corning) coated plates in NPC media of Neurobasal™ (Thermo Fisher, Waltham, MA, USA), 1× GlutaMAX (Gibco, Waltham, MA, USA), 1% N2 (Thermo Fisher), 0.5% B27 without vitamin A (Thermo Fisher), 10 ng/mL human recombinant FGF-2, and 1× antibiotic-antimycotic (Gibco, Waltham, MA, USA). An amount of 10 μm ROCK inhibitor (Y-27632 dihydrochloride, Tocris) was added to media during passaging with TrypLE Express (Thermo Fisher, Waltham, MA, USA) and washed out the subsequent day. Reporter NPCs were passaged once 100% confluent and were discarded after 35 passages from iPSC differentiation. Detailed methods are described in [38].

### 2.3. Immunocytochemistry and Fluorescence Imaging

After 4 days in vitro, NPCs were fixed in 4% paraformaldehyde for 15 min at room temperature. Following three washes with 1× PBS, cells were permeabilized and blocked with blocking reagent (3% normal goat serum (NGS), 0.1% Triton X-100, in 1× PBS) for 30 min at room temperature. Primary antibodies (Rabbit anti-Nestin 1:1500 Abcam #5968) were applied in 3% NGS in 1× PBS overnight at 4 °C in a humidified chamber. Following three 10 min washes with 1× PBS, secondary antibodies (goat anti-rabbit Alexa flour 546, Thermo Fisher) were applied in 3% NGS in 1× PBS for 60 min at room temperature. Coverslips were washed in 1× PBS three times for 10 min with 1 μM DAPI added during the second wash to counterstain cell nuclei. Images were acquired using an AxioImagerZ2 ApoTome confocal microscope (Zeiss, Oberkochen, Germany).

### 2.4. High Throughput Drug Screen

Detailed methods for the development and optimization of the drug screen assays have been described previously [30,38]. 5-Aza-DC is currently the most effective known small molecule treatment to reactivate *FMR1* in FXS cells [32] and was used as the positive control condition for all the present experiments. For the present high throughput (HT) drug screen experiments, confluent reporter NPCs were dissociated and prepared as a single cell suspension. Immediately prior to plating, HT screen media was prepared by adding 10 μM ROCK inhibitor and 0.05 mg/mL Matrigel to cold NPC media. The single-cell suspension of reporter NPCs was added to cold HT screen media to a final concentration of 1500–3000 cells per 5 μL and was immediately plated onto clear-bottom 1536-well tissue culture plates (Corning, Corning, NY, USA). The cell number was adjusted slightly to obtain a consistent Nano-Glo response since different passages of the reporter line proliferated at slightly different rates. After 16 h, the test compounds, 0.1% DMSO (negative control), and 0.3 μM 5-aza-2′-deoxycytidine (5-Aza-DC, positive control) were added to screening plates at the nL scale using acoustic liquid transfer with the Echo 550 liquid-handling instrument (Labcyte, San Jose, CA, USA). NPCs were incubated with test molecules for an additional 72 h before being assayed. *FMR1* reporter expression was measured using the Nano-Glo® Luciferase Assay System (Promega, Madison, WI, USA) according to manufacturer instructions. Luciferase activity was measured on a PheraStar FS plate reader (BMG LABTECH, Ortenberg, Germany).

### 2.5. Cell Viability Assay

Cell viability was measured using the CellTiter-Glo® Luminescent Cell Viability Assay (Promega, Madison, WI, USA) according to manufacturer instructions. Luminescence was measured using a PheraStar FS plate reader.

### 2.6. Small Molecules

The NIH’s Molecular Libraries Probe Production Centers Network (MLPCN) small molecule screening library was used for the main primary screen. The MLPCN library contains over 320,000 compounds and was compiled for large-scale screening projects across a wide range of biological areas (National Center for Biotechnology Information, 2010). No compounds in the MLPCN library have been shown to affect *FMR1* expression in previous studies.

A total of 135 candidate molecules were obtained from commercial vendors (Appendix A).

### 2.7. Data Analysis

For primary and secondary screens, Z’ (formula below) was used as an indicator of assay quality for each test plate.
σ=standard deviation, μ=mean
$$Z^{\prime }=1-3*\frac{\left(\sigma PositiveCtl+\sigma NegativeCtl\right)}{\left|\mu PositiveCtl-\mu NegativeCtl\right|}$$

Assay Relative Light Unit (RLU) values were converted to % mean plate positive control. Response cut-offs to identify “hits” were 10% and 5% of the mean positive control response for the primary and secondary screens, respectively.

For confirmatory testing experiments, one-way analyses of variance were performed at each time point or concentration as an omnibus test of differences in mean response across all treatment conditions. Tukey’s honest significance tests were subsequently run to assess mean differences between each individual pair of treatment conditions. When quantifying differences in treatment group means, the Tukey multiple comparison of means and 95% confidence family-wise confidence level are reported. A critical α of 0.05 defined statistical significance, and all statistical tests were two-sided unless otherwise noted. Statistical analyses were conducted in R (Version 3.5.1) [41].

## 3. Results

### 3.1. Primary Screen of Compounds for Reactivation at the FMR1 Locus in FXS NPCs

FXS NPCs (Figure 2A) were used in the primary screen (Figure 2B) to test a total of 320,587 unique small molecules for reporter activity. Each molecule was tested singularly at a concentration of 12.5 μM, and plate Z’ was used as a robust quality control measure. A quality threshold of 0.20 plate Z’ (mean (min, SD) plate Z’ for entire primary screen = 0.49 (0.20, 0.13)) and response threshold of 10% mean plate 5-Aza-DC reporter activity were used to identify candidate molecules for follow-up testing (Figure 2C). This response threshold was chosen to identify a feasible number of candidate molecules for secondary screening while providing reasonable protection against false positives. Since the screen was for activation, the chance of false-positive results was significantly lower than would be the case for inhibitor screens. As such, the modest Z’ cutoff of 0.2 was deemed acceptable. This activity threshold was equivalent to 6.7 SD above the mean DMSO response across all primary screening plates. A total of 287 unique novel small molecules (0.09% of all molecules tested) exceeded quality and response thresholds and were considered for the confirmatory validation screen.

### 3.2. Selection of Candidate Molecules

Candidate molecules were next selected for the confirmatory validation screen (Figure 3). All validation experiments were carried out in triplicate. All 287 molecules identified in the primary screen were first screened for pan-assay interference [42] using the Free ADME-Tox Filtering Tool (FAFDrugs4) [43]. Eleven molecules were flagged for likely assay interference and excluded from subsequent analysis. Of the remaining 276 molecules, 135 candidate molecules were located from commercial vendors (Appendix A) and included in the secondary screen.

The confirmatory screen procedures are illustrated (Figure 4A). Molecules were tested in triplicate on a 12-point dose-response scale from 0.005 to 20 μM, and *FMR1* reporter responses were compared with the maximal positive control (5-Aza-DC) response (Figure 4B).

Of the 135 candidate molecules tested in the secondary validation screen (Figure 4), two exceeded the response threshold and were used in further validation experiments (Appendix A). Both compounds initially exerted an expected dose-response of *FMR1* reporter activity without cell toxicity. We purchased small amounts of each of these candidate compounds from the original vendor for the MLPCN library, and we were unable to reproduce the results, despite the expected responses from control compounds and the original test compounds in the same assay. Because we were unable to reproduce the initial results with subsequent batches of the compounds, we could not validate these compounds as potential candidates for *FMR1* reactivation strategies.

### 3.3. Confirmatory Validation (Secondary) Screen for Candidate Molecules

All molecules were counter-screened for toxicity using an NPC viability assay (Figure 5). Many test molecules, including 5-Aza-DC, began to show dose-dependent decreases in cell viability at concentrations of ~2–5 μm and higher.

Comparing the dose-dependent cell viability and *FMR1* reporter activity for secondary screen molecules demonstrated that the common dose-dependent decrease in viability was not accompanied by concomitant changes in *FMR1* reporter activity (Figure 6). An *FMR1* reporter response threshold of 5% positive control activity was used to identify novel candidate molecules for further analysis and future validation. Many secondary screen molecules showed a dose-dependent decrease in cell viability but not an *FMR1* reporter response.

## 4. Discussion

### 4.1. Summary and Limitations of Drug Screen

This study provides proof of principle that FXS patient-derived reporter neural cells are a viable research platform to conduct large-scale untargeted screens and a reference for other researchers interested in small molecule drug discovery for FXS. We used human FXS iPSC-derived neural reporter cells to screen over 320,000 unique small molecules for reactivation of *FMR1*, the largest untargeted screen for chemical reactivators of *FMR1* published to date.

In this report, we demonstrated the use of an FXS patient stem cell-based platform to conduct a complete untargeted drug screen for small molecule reactivators of *FMR1* from discovery through confirmatory assays. In an initial untargeted primary screen using a relatively low response threshold to minimize false negatives, 0.09% of all molecules screened exhibited sufficient activity to be considered as candidate molecules for confirmatory screening (Figure 2C). A number of test molecules exhibited dose-dependent decreases in cell viability above concentrations of 2 μM (Figure 5), although this did not coincide with notable decreases in *FMR1* reporter activity (Figure 6). Together, this suggests that the limited number of candidate molecules showing *FMR1* reporter activity was due to a lack of efficacy rather than a function of toxicity and reduced cell viability.

Important limitations of the present experiments should be noted. Although quality-control measures were designed to avoid false negatives in the primary drug screen, the logistical limitations of testing the large library dictated that most screening molecules were only assayed singularly and at a single concentration. This raises the possibility that some molecules that screened negative might have *FMR1* reactivation activity at more optimal concentrations. In addition, this library, although large, is limited to molecules with known biological functions. It will be important to test custom-made libraries in the future.

Reproducibility (batch-dependent activity difference) remains a major consideration. The inability to reproduce results with different batches of compounds could be due to contaminants in initial library compounds, degradation of active compounds, or other unknown reasons.

### 4.2. Validation Assays of Candidate Molecules

Once candidate molecules are identified, several additional assays need to be performed to validate that, in this case, the compounds can reliably elicit *FMR1* reporter activity and lend confidence that the HT screening method did successfully identify true hits. Targeted validation experiments include dose-response assessments and temporal dynamics of potential candidate molecules on *FMR1* reporter activity and cell viability.

Drug discovery for neurological disease often requires that the candidate compounds exert their effects in post-mitotic neurons, rather than proliferating NPCs. To determine whether post-mitotic neurons could be used for validation experiments, testing can be repeated on FXS iPSC-derived reporter neurons.

Further validation experiments may include reactivation in non-reporter (e.g., patient-derived) cell lines as well as studies to uncover the epigenetic and potentially non-epigenetic mechanisms triggered by candidate compounds. In addition, it is important to assess the effects of candidate compounds on non-FXS lines to rule out the possibility that these cells will have different responses.

## Figures and Tables

**Figure 1 cells-11-00069-f001:**
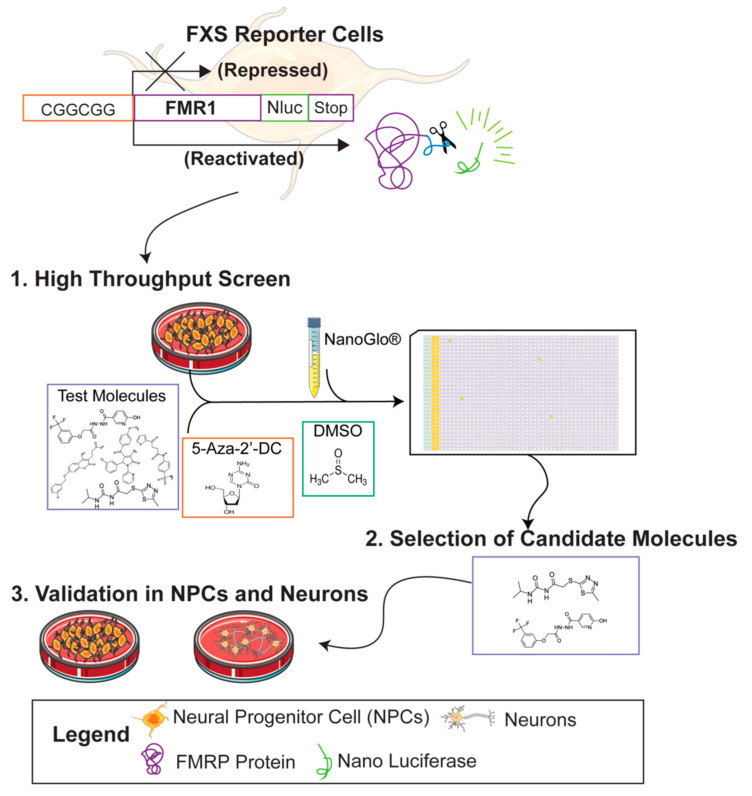
Graphical abstract for drug screen to identify reactivators of the *FMR1* gene.

**Figure 2 cells-11-00069-f002:**
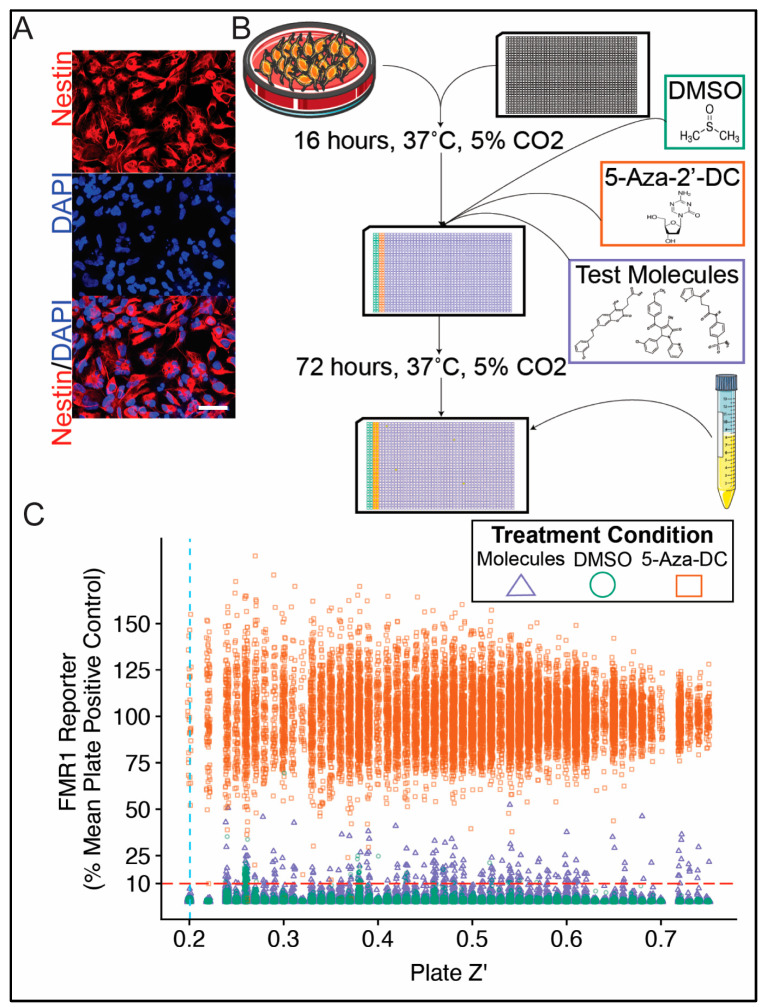
Primary screen for small molecules capable of reactivating *FMR1* in human neural stem cells. Neural progenitor cells (NPCs) were differentiated from human FXS iPSC reporter line. Micrograph of reporter NPCs demonstrates characteristic morphology and expression of Nestin via immunofluorescence (**A**). A high-throughput small molecule screening assay was designed and optimized to test *FMR1* reporter activity in neural progenitor cells (NPCs) (**B**). NPCs were treated with either test compound (320,587 unique compounds, 12.5 µM), 5-aza-2′-deoxycytidine (5-Aza-DC, positive control), or DMSO (vehicle) for 72 h prior to being assayed. The results of the entire primary screen are plotted displaying test plate Z’ on the *x*-axis against individual test molecule response on the *y*-axis (**C**). Molecules were considered as candidates for secondary screening if they exceeded a plate Z’ threshold of 0.20 (blue dashed vertical line) and an individual response threshold of 10% mean plate 5-Aza-DC response (red dashed horizontal line). Scale bar = 50 μm.

**Figure 3 cells-11-00069-f003:**
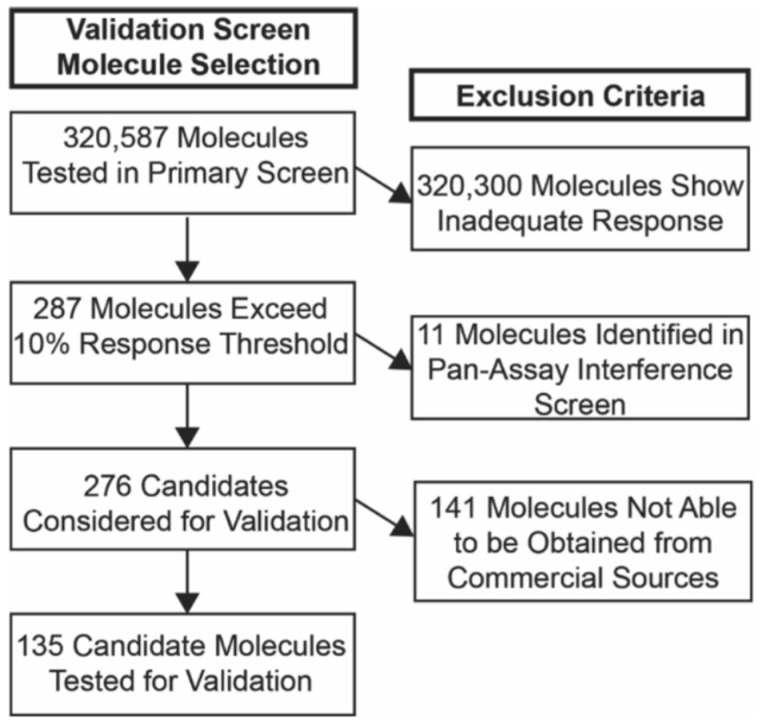
Selection of candidate molecules for secondary screen. Flow diagram shows process for selecting candidates for validation experiments.

**Figure 4 cells-11-00069-f004:**
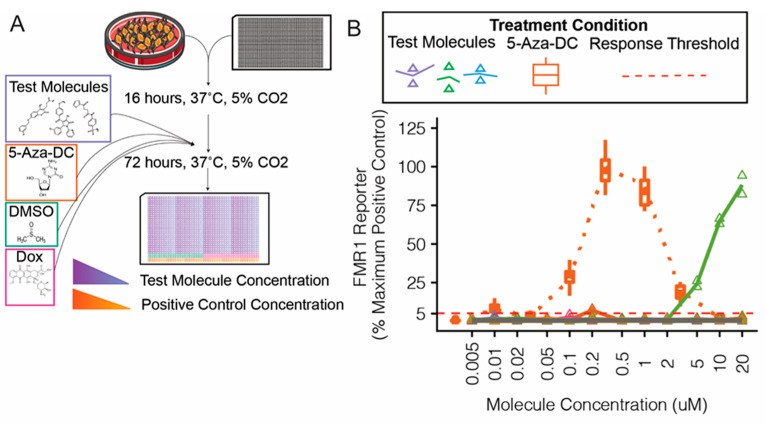
Confirmatory validation screen of candidate molecules for *FMR1* reporter activity in neural progenitor cells. Candidate small molecules were selected for secondary screening on the basis of response in primary screen and availability (**A**). Reporter neural progenitor cells (NPCs) were treated for 72 h with candidate molecules from the MLPCN library, 5-aza-2′-deoxycytidine (5-Aza-DC, positive control, orange boxes and line), or DMSO (vehicle, gray triangles, and line) (**B**).

**Figure 5 cells-11-00069-f005:**
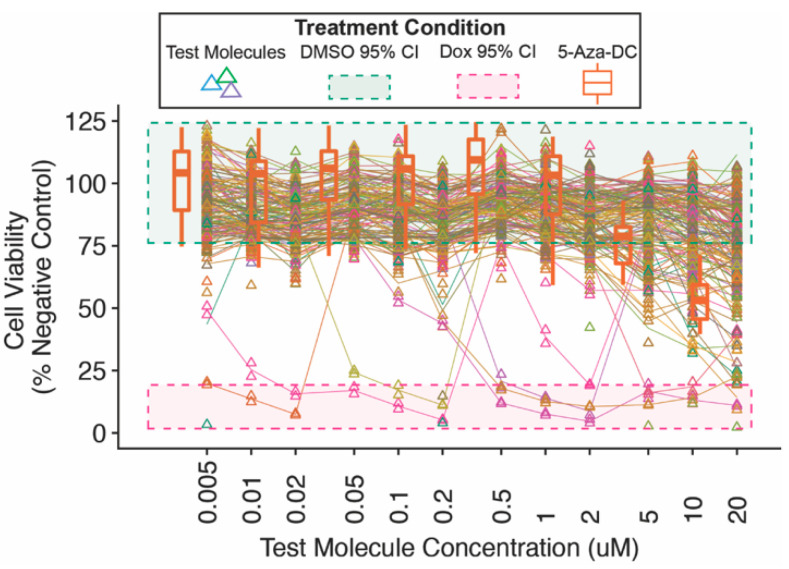
Secondary screen neural progenitor cell viability. Reporter neural progenitor cells (NPCs) were treated for 72 h with candidate molecules, 5-aza-2′-deoxycytidine (5-Aza-DC, positive control) or doxorubicin (kill control), or DMSO (vehicle). NPCs treated with each concentration of candidate molecule were counter-screened for cell viability.

**Figure 6 cells-11-00069-f006:**
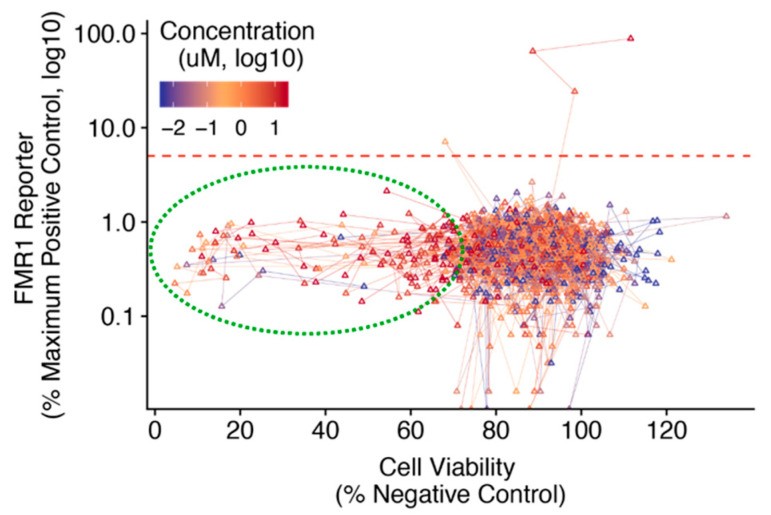
Confirmatory validation screen of candidate molecule dose-response characteristics of both *FMR1* reporter response and cell viability in FXS neural progenitor cells. FXS reporter neural progenitor cells (NPCs) were treated with secondary screen candidate molecules for 72 h. Mean cell viability (*x*-axis) is plotted against mean *FMR1* reporter activity (*y*-axis) across a range of treatment concentrations (color scale). Results from each unique test molecule are connected with lines in order of increasing concentration. A response threshold of 5% mean 5-Aza-DC (positive control) activity was used to identify candidate molecules for follow-up analysis (red dashed line).

## Data Availability

Not applicable.

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
