# Peer review of "High Throughput Small Molecule Screen for Reactivation of FMR1 in Fragile X Syndrome Human Neural Cells"

_cells, 2021, doi:10.3390/cells11010069_

Round 1
Reviewer 1 Report
The authors have revised the manuscript to my satisfaction.
Please check for minor spelling errors and italicize FMR1 in the text.
Author Response
The authors have revised the manuscript to my satisfaction.
Please check for minor spelling errors and italicize FMR1 in the text.
We have corrected minor spelling errors and italicized FMR1. We have also changed the protein name from FMRP to FMR1, the new official name of this protein (lines 43-44).
Reviewer 2 Report
In this manuscript by Hunt et al, they provide the results of a large small molecule screen in human FX NPCs using a nanoluciferase tagged FMRP protein in cells where the FX locus in transcriptionally silenced by the CGG repeat. The screen had a solid design with a good z' score and an appropriate positive control. It led to identification of about 300 molecules for further analysis. Unfortunately, none of these led to robust and reproducible reactivation of FMRP expression in this system. Despite the negative result, I think the findings are timely and important. I have a few questions that should be addressed by the authors.
1) Can you address whether the nanoluciferase tag is present on all FMR1 isoforms or only some isoforms? If the latter, this should be noted as a limitation of the approach.
2) The screen was done in NPCs rather than differentiated neurons- presumably because of the large size of the screen. Can the authors comment on the feasibility of such a screen in neurons and what limitations the use of NPCs places on the screen (given that it is thought that loss of FMRP in mature neurons is a problem in FXS).
3) 5-aza-dC leads to a transient reactivation of FMR1 transcription that does not persist, but which does create a more open chromatin state. Did the authors consider testing their primary hits in NPCs after they were first treated with 5-aza-dC to see if they could maintain expression of FMRP in this setting?
4) It is unfortunate that the two most promising "hits" in the screen did not reproduce when the molecules were re-obtained. The authors did not mention which molecules these were. Is there any method to retroactively identify whether those molecules in the original "hit" solutions are indeed the molecule you thought they were or perhaps were something else? (assuming that is some modification or contaminant)? Perhaps it is some oxidized version of the re-obtained molecule? Could be worth looking into given the significant effort put into the project to date to ID those two hits.
Author Response
Please see attachment.
This manuscript is a resubmission of an earlier submission. The following is a list of the peer review reports and author responses from that submission.
Round 1
Reviewer 1 Report
Overall:
This article did not provide a lot of results. They only showed the results of a 2-step screening process, from which they identified 2 potential candidates. They did not provide further insight on to the validation of these 2 compounds. In my opinion, this article should be reorganized into a method paper rather than a research one, as it did not provide enough data for the scientific community.
Introduction:
Nothing to say on that section. The introduction contains all the information required to understand the context of the study.
Material and Methods:
FMR1-Nano Luciferase Reporter Cell Lines
The authors should mention the passaging procedure used for maintaining the cell line.
Line 114
The text doesn’t need to be bold
Line 122
The authors should mention the secondary antibody used.
Line 140-141
The authors should mention which parameters and how they were optimized.
Results:
Figure 2A
Why the authors only decided to use Nestin as an NPC marker? They should, at least, show another marker such as PAX6.
There is also no scale bar.
Line 201
The authors mention that all validation experiments were carried out in triplicate, but latter they state that all molecules were tested in duplicates (line 208)
Line 208-2011
The results obtained from the dose-response validation experiment are not clearly mentioned. How many compounds showed reporter activity greater than the threshold?
Reviewer 2 Report
The authors have revised the manuscript to address my concerns with the previous version. I have the following comments for the revised manuscript:
Major concerns:
- The results of the confirmatory validation (secondary) screen are not presented clearly in the Results section. Please provide this information which is partly included in the legend of figure 6 and partly in the Discussion section. It is not clear if the results shown in figure 4 were obtained with the original compounds from the MLPCN library or with commercial compounds. The authors need to clearly state in the Results section that the two candidate compounds identified in the primary screen with MLPCN library failed in the confirmatory validation screen done with compounds obtained from a commercial source.
- The authors could include the resynthesis data only if they compared the structure of resynthesized compounds with that of the original compounds in the MLPCN library. Because both the resynthesized compounds and those obtained from the commercial source could not be validated in the confirmatory screens, it is not clear what this data adds to the paper without comparing it to the structure of the original compound. Therefore, the paragraph describing the resynthesis of compounds (lines 278-282) and supplementary figure 1 need not be included and removed from the Discussion.
- Similarly, given that the manuscript has been modified to conclude no active compounds were identified in the screen that could be validated, perhaps there is no need for the paragraph describing the hemolytic activity of the 2 compounds (lines 283-287) and no need for supplementary figures 2 and 3. These should also be removed from the manuscript.
- The authors have deleted all the information about HTS using neurons in this version of the manuscript. Therefore, the sentence in lines 69-73 needs to be rephrased as it seems to suggest that the current screen was done using neurons.
- Please consider rewriting the Discussion section after making changes suggested above to highlight the importance of the current study as a proof-of principle study for very large HTS using reporter cell lines, and the limitations of the current study.
Minor edits:
- Figure 2 legend, line 194- please correct the symbol of µM.
- Figure 4 legend, line 216-please mention what color represents doxorubicin data in Panel B, remove the reference in the legend if data not shown. Similarly, the data for DMSO (in gray) is not clearly visible in 4B.